# FEC-Real: A KAN based Model for Improving Investment Strategies

## Abstract

The application of mathematical and computational techniques in financial investment has emerged as a prominent area of research, leading to the development of various tasks including factor mining, stock prediction, and analysis of financial statements. In this work we particularly focus on the task of predicting the future trend for stocks. In existing fintech research different transformer-based models have been explored for predicting future stock trend. This study is motivated by the need for a more efficient network architecture that can enhance the interpretation of real-time data. However, transformer based models are not always efficient for real world high speed trading data. To address this, we particularly explore the effectiveness of Kolmogorov Arnold Network (KAN) for financial time series model. We propose a KAN based encoder (FTS2K) which utilizes both KAN and transformer architecture to predict future stock price movements. Empirical results show that our proposed Encoder improves yields an average accuracy enhancement of 2.62% across state-of-the-art (SOTA) time series models. Our approach consistently outperforms in four datasets (i.e. China A Daily, China A Min, China Futures Min, Dow 12 Daily), achieving superior results in both ACC and Top-100 ACC metrics.

## 1 Introduction

Predicting stock price movements has consistently garnered significant interest from both academic and finance industry. Within the global economic environment, the increasing uncertainty of financial markets impose greater demands on investors and market analysts Raddant & Kenett (2021). Financial time series modeling, as a crucial technical tool, assists in extracting valuable information from historical data and predicting future market trends, has played a pivotal role in the domain of modern finance Zakhidov (2024). By leveraging high-accracy market forecasts, investors can develop investment strategies more scientifically. This includes increasing holdings in anticipation of a market rise, and reducing holdings or seeking hedging strategies in anticipation of a market decline, thereby optimizing investment returns Navon & Keller (2017).

Traditional statistical methods for financial modeling, however, often struggle to grasp the complex, dynamic, and nonlinear relationships present in financial data Zhang et al. (2024). With the advent of deep learning, significant advancements have been achieved in financial time series modeling tasks through deep neural networks, models employing Recurrent Neural Network (RNN), Long Short-Term Memory Network (LSTM) , and attention mechanisms have demonstrated their capacity to notably enhance prediction accuracy under certain conditions Sezer et al. (2020). Given the challenge of balancing prediction accuracy and generalization within a single model, researchers have started to investigate hybrid techniques Kim & Won (2018).

In that direction we propose a hybrid model which exploits KAN based encoding for improving predictionLiu et al. (2024b). Current financial time series research is mainly based on historical data, and there are few studies that are tested in the real worldLi et al. (2024). To address these, we propose an encoder combining transformer and KAN in our model. As illustrated in Figure 1, Financial market data, depicted on the left, undergoes encoding via a encoder before being sent to multiple agents for prediction purposes. Subsequently, our hybrid algorithm integrate these predictions, delivering the final forecast for each time series. The key contributions of our work include:

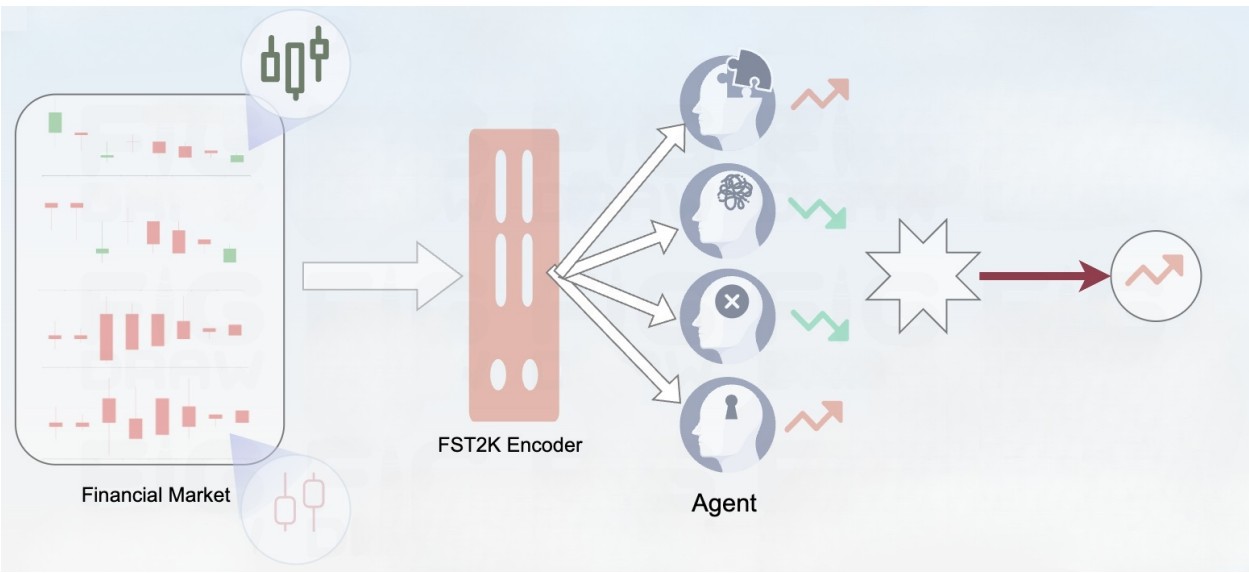

Figure 1: Overview of the FEC-Real trading strategy workflow in real-world trading environment

1. We explore the effectiveness of KAN in financial time series processing.

2. We proposed a financial data encoder named FTS2K which improved the accuracy of the prediction results of other mainstream SOTA time series models by 2.62%.

3. We proposed the FEC-Real hybrid decision model and designed the Top-100 ACC as the evaluation indicator based on practical applications. FEC-Real achieved the best results on four financial datasets. Crucially, we deployed it in a real trading environment and achieved a real excess return of up to 46. 35%, offering valuable information for future financial time series experiments.

## 2 Related Works

In the financial domain, correct forecasting market behavior and price fluctuations is of paramount importance Khedr et al. (2017). The importance of time series analysis in the financial field is not only reflected in market forecasting, but also involves risk management, product pricing, and macroeconomic analysis Stock & Watson (1998); Chen et al. (2024); Lin et al. (2021); Xu et al. (2021).

To effectively integrate time series analysis with the financial domain, it is typically to utilize historical data, encompassing asset transaction prices, trading volumes, economic indicators, and market sentiment indicators Sezer et al. (2020). Traditional methodologies, such as autoregressive integrated moving average (ARIMA) models and generalized autoregressive conditional heteroskedasticity (GARCH) models, have been extensively employed to identify linear and non-linear patterns in financial data Merabet et al. (2021). Nevertheless, these models commonly operate under the assumption of data stationarity and frequently encounter difficulties in adequately representing the intricate dynamics and non-linearities inherent within financial markets Tsay (2005).

Subsequently, scholars increasingly turned their attention to interval time series models, which utilize the maximum and minimum prices of financial assets to form intervals. In contrast to conventional point value models, interval models are capable of capturing a broader spectrum of market information, including the range of price fluctuations, thereby potentially enhancing the accuracy of forecasts Moore et al. (2009).

With the advent of machine learning technology, methodologies such as the support vector machine (SVM) Tay & Cao (2001) and random forest (RF) Khaidem et al. (2016) have been increasingly integrated into the domain of financial time series forecasting Kumar & Thenmozhi (2006). In recent years, there is a growing

focus on deep learning methods, which are particularly well-suited for the prediction of highly volatile financial market data due to their capacity to capture dependencies within time series Zeng et al. (2024). Notably, the implementation of attention mechanisms has significantly enhanced the model's capability to analyze time series Sawhney et al. (2021); Wu et al. (2022); Nie et al. (2022a); Zhou et al. (2021). Moreover, hybrid models such as ARIMA-LSTM and CNN-LSTM prove to be highly effective in handling complex financial time series data Choi (2018); Widiputra et al. (2021). These hybrid methodologies have led to substantial improvements in the accuracy of forecasts.

Kolmogorov-Arnold Networks (KAN) represents a novel neural network architecture that draws inspiration from the Kolmogorov-Arnold representation theorem Liu et al. (2024b). According to this theorem, any multivariable continuous function can be decomposed into a combination of a finite number of single-variable continuous functions Kolmogorov (1961). KAN fundamentally obviates the need for reliance on linear weight matrices by substituting fixed activation functions with learnable ones Cheon (2024). Such an framework design provides substantial flexibility and facilitates the simulation of complex functions with a reduced number of parameters, thereby enhancing the model's interpretability Genet & Inzirillo (2024). Models based on KAN are adept at adapting to nonlinear variations in time series data through the acquisition of activation functions, rendering them particularly effective in capturing intricate temporal dependencies and yielding more accuracy prediction results Vaca-Rubio et al. (2024).

## 3 Methodology

The architecture of the proposed model Financial Encoder Classification for Real Environment (FEC-Real) includes three principal components: a) an encoder, b) a classifier to handle the encoded data, and c) a voting method that multiple models in a hybrid manner. Each one of them is described as follows.

### 3.1 FTS2K Encoder

Existing research Sagheer & Kotb (2019); Liu et al. (2024a) have shown that the accuracy of time series tasks can be improved by encoding them with a pre-trained encoder. Consequently, we design an encoder named **FTS2K** that utilizes a wide variety of financial trading data with different time series lengths to effectively capture features within financial asset trading data.

Figure 2 illustrates the architecture of the FTS2K encoder, which initially processes the input data, integrates positional encoding, and subsequently channels it through the attention layer followed by a feedforward neural network. We used attention layer since numerous experiments and time series models have demonstrated that incorporating attention mechanisms significantly enhance the effectiveness of extraction in time series data, thereby markedly improving classification accuracy Abbasimehr & Paki (2022); Fang et al. (2023); Wang et al. (2020); Niu et al. (2024). After processing by the feedforward neural network, the output is further refined through two Multilayer Perceptron (MLP) layers, serving as a compression mechanism, to ultimately produce encoded information of 20 dimensions.

Existing study has shown the effectiveness of KAN network for time series forecasting. However, the efficiency of the KAN model is affected in high dimensional data Han et al. (2024). Furthermore, the training time for high dimensional data is very slow in KAN netwroks. Consequently, the FTS2K encoder has been devised as a module to integrate conventional neural networks with the KAN network in a low dimensional space, thereby facilitating the improved extraction of latent features in extensive financial sequences and reducing the dimensionality of such sequences to optimally align with the KAN classification network's operational efficacy.

To enhance the performance and generalization capability of the FTS2K encoder, we have implemented several critical strategies during its training phase: (1) Diversification of training data: We employed a variety of financial time series data, encompassing different lengths and asset categories for training. This includes both high-frequency and low-frequency data from diverse financial instruments such as stocks and futures. Our goal is to enable the FTS2K encoder to capture more universal features of financial time series by broadening the data diversity. (2) Dynamic sequence lengths: We dynamically adjust the lengths of the input sequences throughout the training process. This approach not only improves the model's proficiency

in processing sequences of varying lengths, but also improves its ability to recognize long-term dependencies. The data used ranged from 30 minutes to 1 year in duration. (3) High-to-low dimensional data conversion: While most encoders maintain data in high dimensions Chen & Guo (2023), processing high-dimensional data remains a challenge for KAN. The FTS2K encoder overcomes this by producing low-dimensional data, which KAN processes more efficiently, through a two-layer neural network.

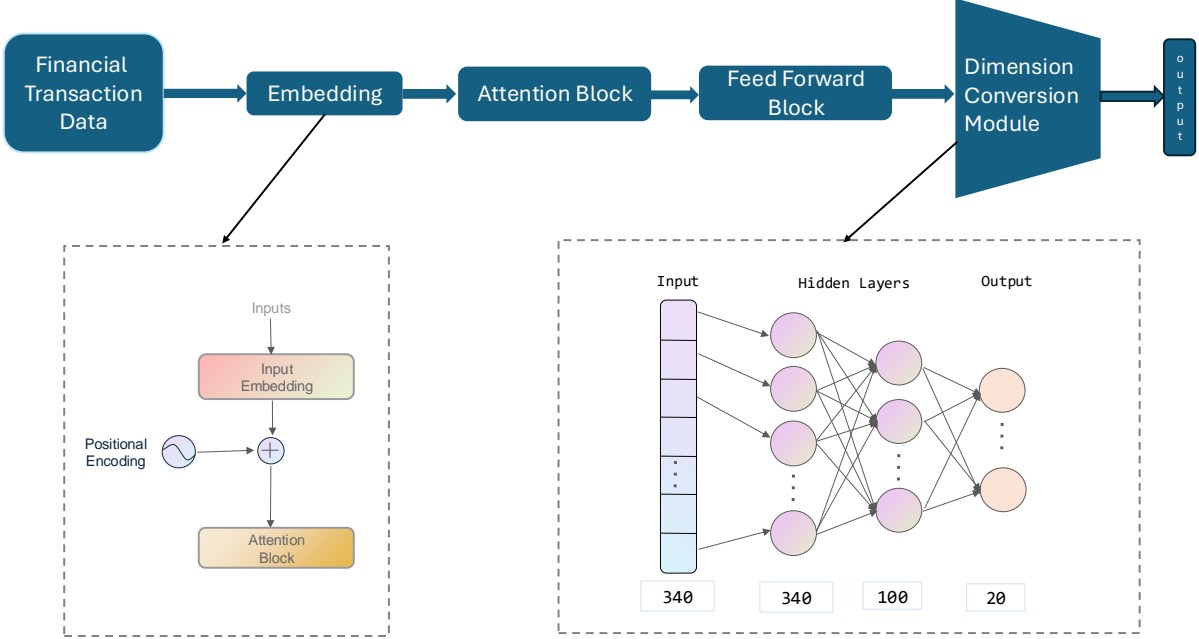

Figure 2: The overall architecture of FTS2K Encoder.

## 3.2 Classification Model

In our task, we use the KAN network with three hidden layers as a classification network for the future trend of financial time series. Specifically, the financial time series will be input into the KAN network, and after passing through three hidden layers, there are two output nodes to judge the future trend of the financial time series. The concept of the KAN network structure is derived from the Kolmogorov-Arnold representation theorem, which states that any multivariate continuous function $f$ can be represented as a composition of univariate functions $\{\phi_{q,p}\}$ and the addition operation (as shown in Equation 1). $\{\phi_{q,p}\}$ map each input variable $x_p$ such $\Phi_q : \mathbb{R} \to \mathbb{R}$ and $\phi_{q,p} : [0,1] \to \mathbb{R}$.

$$f(x_1, \ldots, x_n) = \sum_{q=1}^{2n+1} \Phi_q \left( \sum_{p=1}^{n} \phi_{q,p}(x_p) \right) \tag{1}$$

A KAN layer can be defined as shown in Equation 2.

$$\Phi = \phi_{q,p} \tag{2}$$

As shown in Equation 2, $\{\phi_{q,p}(\cdot)\}$ with $q = 1, \ldots, N_{in}$ and $p = 1, \ldots, N_{out}$, where $N_{in}$ and $N_{out}$ denote the number of inputs and the number of outputs, respectively, and $\phi_{q,p}$ are the trainable spline functions described above. A generic deeper KAN can be expressed by the composition $L$ layers as shown in Equation 3.

$$\mathbf{y} = \text{KAN}(\mathbf{x}) = (\mathbf{\Phi}_L \circ \mathbf{\Phi}_{L-1} \circ \ldots \circ \mathbf{\Phi}_1)(\mathbf{x}) \tag{3}$$

Table 1: Dataset Details for Experiment Setup

| Data Set | Encoder Training | | Forecasting Task Training | | Forecasting Task Test | |
|---|---|---|---|---|---|---|
| | Time Range | Number | Time Range | Number | Time Range | Number |
| A Daily | 2022-06-20 to 2024-03-19 | 1,577,928 | 2024-03-20 to 2024-05-15 | 81,098 | 2024-05-16 to 2024-06-20 | 33,591 |
| A Min | 2022-06-20 to 2024-03-19 | 1,192,117 | 2024-03-20 to 2024-05-15 | 120,951 | 2024-05-16 to 2024-06-20 | 24,819 |
| Futures | 2023-06-20 to 2024-03-19 | 151,560 | 2024-03-20 to 2024-05-15 | 32,195 | 2024-05-16 to 2024-06-20 | 9,622 |
| Dow12 | 2018-07-01 to 2020-07-01 | 12,278 | 2020-07-02 to 2020-12-31 | 4,928 | 2021-01-01 to 2021-03-01 | 989 |

## 3.3 Multi-Agent Model

Due to the significant impact of stock price forecasts on investment decisions, there is a high demand for robustness in their predictions. Therefore, utmost accuracy is sought in forecasting. To enhance the credibility of the predictions, a multi-agent approach is devised. Specifically, we integrate the predictions of various time series models (KAN,Transformer, TimesNet, PatchTST, Informer) based on the dataset. By employing a method of setting confidence threshold, we eliminate forecasts with low confidence and conduct voting among the high-confidence results.

$$p_i = m_i(x), \quad i = 1, 2, \ldots, n \tag{4}$$

$$N_0 = \sum_{i=1}^{n} 1_{p_i=0, c_i>0.8} \tag{5}$$

$$N_1 = \sum_{i=1}^{n} 1_{p_i=1, c_i>0.8} \tag{6}$$

$$P_{final}(x) = \begin{cases} 0 & \text{if } N_0 > N_1 \\ p_i, (MAX(c_i)) & \text{if } N_0 = N_1 \\ 1 & \text{otherwise} \end{cases} \tag{7}$$

In Equation 4, $x$ represents the financial time series, $m_i(x)$ represents the numuber $i$ time series model, $p_i$ represents the prediction result of the numuber $i$ model, $n$ represents the number of models, $N_0$ in Equation 5 represents the number of models in the model set that predict $x$ as negative, and $c_i$ represents the confidence of the prediction result of the model. $N_1$ in Equation 6 represents the number of models in the model set that predict $x$ as positive. In Equation 7, $P_{final(x)}$ represents the final prediction result of time series $x$. When $N_0$ is greater than $N_1$, the prediction result is negative. When $N_0$ is equal to $N_1$, the result with the highest confidence in the model set is selected as the final result. When $N_0$ is less than $N_1$, the prediction result is positive.

# 4 EXPERIMENT

Here we describe our experimental datasets, evaluation metrics, and baseline models.

## 4.1 Datasets and Processing

We evaluate our model on four diverse financial datasets:

**China A Daily:** We utilize daily trading data of $5,364$ listed companies on China's A-share market, covering a comprehensive two-year period from June 20, 2020, to June 20, 2024. We crawl the dataset from a publicly available website[1]. This dataset encompasses daily closing prices and trading volumes. By constructing time series data across different intervals: 10 days, 20 days, and 30 days—and subsequent to the removal of

---
[1] https://www.tushare.pro/

anomalies and irrelevant data, we successfully curate a dataset comprising 1.5 million instances of A-share daily trading time series data.

**China A Min:** We also leverage minute-level trading data from 5,364 listed companies on the Chinese A-Shares market, covering a period from June 20, 2020, to June 20, 2024. We collect per-minute prices and trading volumes at various time intervals, specifically 30 minutes, 90 minutes, and 180 minutes. After the exclusion of exceptional and irrelevant data, we compile a dataset comprising one million time series data points for analysis. The objective of this dataset is to unearth insights into the minute-level trading dynamics and characteristics prevalent in the Chinese A-Shares market.

**China Futures Min:** This dataset comprises minute-level trading data for five main commodity futures (i.e, Soda ash, Rebar, Crude, Copper, Gold) in China, covering the period from June 20, 2023 to June 20, 2024, including Per minute closing prices and trading volumes. Time series data are established for different time intervals of 10 minutes, 30 minutes, and 180 minutes, resulting in a total of 200,000 clean and relevant data points after the removal of anomalies and irrelevant information.

**Dow12 Daily:** The dataset includes daily trading data for 12 stocks listed in the Dow Jones Index[2], along with investor behavior indicators for each stock extracted from news text data Zhang et al. (2024).

Table 1 given the division of the four datasets used in encoder training and classification tasks.

## 4.2 Implement Details

Experiments are conducted on an Nvidia 4090 GPU. The implementation of those SOTA time series models is based on Time-Series-Library[3].

## 4.3 Evaluation Metrics

We chose **Overall Accuracy (ACC)** and **Top-X Confidence Accuracy (T-X ACC)** as our evaluation metrics for evaluating model performance.

ACC is a measure reflecting the proportion of predictions that match the actual outcomes in the test set. Specifically, it is defined as:

$$\text{ACC} = \frac{N_{correct}}{N_{total}}, \tag{8}$$

$N_{correct}$ represent the number of correct predictions, $N_{total}$ represent the total number of predictions.

**T-X ACC** pertains to the accuracy among the predictions with the highest confidence levels, specifically, the top $X$ ($X \in \mathbb{N}_{>0}$) confidence score results of prediction . It essentially measures the proportion of these high-confidence predictions that are correct, and is given by:

$$\text{T-X ACC} = \frac{N_{top}}{X}, \tag{9}$$

$N_{top}$ represent the number of correct predictions among top X confident predictions.

## 4.4 Baselines

The different baselines used in our experiment setup are described as follows.

**Transformer (Vaswani (2017)):** In this approach we use the encoding obtained from the transformer model to build the classifier.

**KAN Liu et al. (2024b):** KANs is a novel neural network architecture that enhances network performance and interpretability by replacing weight parameters with learnable univariate functions.

---

[2]https://en.wikipedia.org/wiki/Dow_Jones_Industrial_Average
[3]www.github.com/thuml/Time-series-Library

Table 2: Experimental results of time series models on low-dimensional data. The letter 'E' following the dataset's name signifies that the dataset has been encoded by FTS2K. The best performing method for each metric is highlighted in bold font, while the second-best-performing method are underlined.

| Model | China A Daily | | China A Daily E | | Dow12 Daily | | Dow12 Daily E | |
|---|---|---|---|---|---|---|---|---|
| | ACC ↑ | Top-100 ACC ↑ | ACC ↑ | Top-100 ACC ↑ | ACC ↑ | Top-100 ACC ↑ | ACC ↑ | Top-100 ACC ↑ |
| FEC-Real (Our) | **0.5894** | 0.62 | **0.5837** | 0.60 | **0.6148** | 0.61 | **0.6127** | 0.62 |
| Transformer Vaswani (2017) | 0.5664 | 0.58 | 0.5770 | 0.58 | 0.5814 | 0.60 | 0.5804 | 0.60 |
| Informer Zhou et al. (2021) | 0.5637 | 0.55 | 0.5698 | 0.57 | 0.5885 | 0.60 | 0.5794 | 0.59 |
| PatchTST Nie et al. (2022b) | 0.5496 | 0.59 | 0.5625 | 0.57 | 0.5713 | 0.57 | 0.5581 | 0.58 |
| TimesNet Wu et al. (2022) | 0.5509 | 0.54 | 0.5792 | 0.58 | 0.5429 | 0.57 | 0.5642 | 0.58 |
| KAN Liu et al. (2024b) | 0.5711 | 0.60 | 0.5744 | 0.60 | 0.5814 | 0.60 | 0.5905 | 0.60 |
| MLP Yu et al. (2024) | 0.5392 | 0.53 | 0.5602 | 0.55 | 0.5490 | 0.57 | 0.5551 | 0.57 |
| TimeMixer Wang et al. (2024) | 0.5541 | 0.55 | 0.5653 | 0.58 | 0.5632 | 0.57 | 0.5652 | 0.59 |
| Mamba Gu & Dao (2023) | 0.5661 | 0.56 | 0.5685 | 0.57 | 0.5591 | 0.56 | 0.5561 | 0.58 |
| XGBoost Chen & Guestrin (2016) | 0.5490 | 0.54 | 0.5547 | 0.56 | 0.5536 | 0.56 | 0.5782 | 0.58 |
| LSTM Graves & Graves (2012) | 0.5433 | 0.54 | 0.5498 | 0.55 | 0.5496 | 0.55 | 0.5579 | 0.56 |
| MASTER Li et al. (2024) | 0.5657 | 0.56 | 0.5715 | 0.58 | 0.5644 | 0.56 | 0.5889 | 0.59 |

**MLP Yu et al. (2024):** MLP is a type of feedforward artificial neural network. Here we use at least three layers of nodes: an input layer, one or more hidden layers, and an output layer in a feedforward artificial network.

**TimesNet Wu et al. (2022):** TimesNet tackle the limitations of 1D time series in representation capability, extends the analysis of temporal variations into the 2D space by transforming the 1D time series into a set of 2D tensors based on multiple periods.

**Informer Zhou et al. (2021):** Propose a ProbSparse self-attention mechanism, which has comparable performance in the alignment of the dependency of the sequences.

**PatchTST Nie et al. (2022b):** A transformer model based on patches. Segment the time series into multiple small patches for modeling and prediction, and can capture local patterns and long-term dependencies in the time series.

**TimeMixer (Wang et al. (2024)):** The historical information and future predictions are processed respectively through the Past-Decomposable-Mixing (PDM) and Future-Multipredictor-Mixing (FMM) modules.

**Mamba Gu & Dao (2023):** Using the SSM parameters as input functions and integrate that into a simplified end-to-end neural network framework.

**XGBoost Chen & Guestrin (2016):** XGBoost (Extreme Gradient Boosting) is a popular gradient-boosted decision tree algorithm widely used for time series forecasting tasks.

**LSTM Graves & Graves (2012):** LSTM is a type of recurrent neural network designed to model long-term dependencies in sequential data. It uses memory cells and gates to retain or forget information over time, making it effective for time-series forecasting.

**MASTER Li et al. (2024):** The Master represents an innovative stock prediction framework that employs machine learning techniques to comprehensively analyze short-term market dynamics and cross-temporal stock correlations. By innovatively integrating a dynamic gating mechanism, this approach effectively incorporates multidimensional market data to achieve intelligent feature selection and scenario adaptation.

## 5 Results and Analysis

Table 2 and 3 given the experimental results of different models on various financial market datasets (different markets, time dimensions, and types of market information). The tables enumerate multiple models, includ-

Table 3: Experimental results of time series models on high-dimensional data. The letter 'E' following the dataset's name signifies that the dataset has been encoded encoded by FTS2K. The best performing method for each metric is highlighted in bold font, while the second-best-performing method are underlined.

| Model | China-A Min | | China-A Min E | | China-Futures Min | | China-Futures E | |
|---|---|---|---|---|---|---|---|---|
| | ACC ↑ | Top-100 ACC ↑ | ACC ↑ | Top-100 ACC ↑ | ACC ↑ | Top-100 ACC ↑ | ACC ↑ | Top-100 ACC ↑ |
| FEC-Real (Our) | **0.5811** | 0.63 | **0.6072** | 0.64 | **0.6226** | 0.64 | **0.6198** | 0.62 |
| Transformer Vaswani (2017) | 0.5718 | 0.58 | 0.5834 | 0.58 | 0.6073 | 0.64 | 0.5995 | 0.62 |
| Informer Zhou et al. (2021) | 0.5606 | 0.59 | 0.5867 | 0.60 | 0.5724 | 0.61 | 0.5803 | 0.58 |
| PatchTST Nie et al. (2022b) | 0.5578 | 0.57 | 0.5877 | 0.57 | 0.5420 | 0.56 | 0.5792 | 0.59 |
| TimesNet Wu et al. (2022) | 0.5424 | 0.58 | 0.5773 | 0.58 | 0.5801 | 0.58 | 0.5889 | 0.61 |
| KAN Liu et al. (2024b) | 0.5359 | 0.54 | 0.5929 | 0.61 | 0.5573 | 0.57 | 0.5845 | 0.60 |
| MLP Yu et al. (2024) | 0.5487 | 0.58 | 0.5636 | 0.55 | 0.5507 | 0.57 | 0.5776 | 0.59 |
| TimeMixer Wang et al. (2024) | 0.5592 | 0.57 | 0.5779 | 0.58 | 0.5779 | 0.60 | 0.5722 | 0.57 |
| Mamba Gu & Dao (2023) | 0.5637 | 0.58 | 0.5809 | 0.60 | 0.5607 | 0.58 | 0.5831 | 0.58 |
| XGBoost Chen & Guestrin (2016) | 0.5339 | 0.55 | 0.5209 | 0.53 | 0.5690 | 0.56 | 0.5714 | 0.57 |
| LSTM Graves & Graves (2012) | 0.5391 | 0.53 | 0.5415 | 0.53 | 0.5504 | 0.53 | 0.5528 | 0.53 |
| MASTER Li et al. (2024) | 0.5605 | 0.56 | 0.5937 | 0.59 | 0.5840 | 0.57 | 0.5886 | 0.58 |

ing our FEC-Real model (presumably referring to the model proposed by the experimenters themselves) and several state-of-the-art models in the field of time series, such as Transformer, Informer, and PatchTST. Each model is evaluated on four different datasets: China-A Daily (stocks, low dimension), China-A Min (stocks, high dimension), China-Futures Min (futures, high dimension), and Dow12 Daily (stocks, news information). Each dataset is assessed based on two performance metrics: ACC (accuracy) and Top-100 ACC (accuracy of the top 100 predictions). The models in Table 2 directly process the data from the datasets, while the data in Table 3 is encoded by the FTS2K encoder before being fed into the models for prediction.

In Table 2, the experimental results for each model applied to the low-dimensional dataset are given. The FEC-Real model exhibit superior performance across all datasets and metrics. Table 2, KAN attains the highest accuracy of 0.5711 on the relatively low-dimensional China-A Daily dataset. Furthermore, it achieves an accuracy of 0.5814 on the Dow12 Daily low-dimensional dataset, which includes the news index. Employing the FTS2K encoder to process the low-dimensional data resulted in an average increase of 1.001% in prediction accuracy for the China-A Daily dataset and an improvement of 0.1167% for the Dow12 Daily dataset.

Table 3 gives the performance metrics of each model on high-dimensional datasets. Utilizing the FTS2K encoder for high-dimensional data significantly enhances the KAN network's accuracy, with improvements of 5.7% and 2.72% observed on the China-A Min and China-Futures Min datasets, respectively. The FTS2K encoder not only markedly boosts the KAN network's predictive accuracy in complex high-dimensional financial tasks but also elevates the performance of other models on high-dimensional datasets. Specifically, the average prediction accuracy on the China-A Min dataset saw an increase of 2.627%, while the China-Futures Min dataset experienced an average improvement of 1.257%.

By comparing Table 2 and Table 3, it can be found that when the KAN network processes the high-dimensional datasets China-A Min and China-Futures Min, the accuracy drops significantly, only 0.5359 and 0.5573 respectively. Transformer has the best overall performance in the four data sets without using the FTS2K encoder, especially in the two high-dimensional data. The accuracy rate in the two low-dimensional data sets is basically the same as KAN. However, the introduction of the FTS2K encoder makes up for the performance degradation of the KAN network when processing high-dimensional data, allowing KAN to efficiently complete various financial market time series tasks while maintaining a simple network structure. In addition, the performance of the FTS2K encoder in various financial timing tasks has been improved to varying degrees under different models, and it can be widely used in research on financial timing problems.

When compared to the overall accuracy rate, the Top-100 ACC indicator provides greater reference significance for the task of financial asset recommendation. By comparing the ACC and Top-100 ACC indicators, it can be observed that in most cases, the predictive accuracy of the Top-100 is 2-3% higher than the overall

accuracy. Furthermore, our method has achieved the top ranking in all experiments, with accuracy rates exceeding 60% in all cases, which is extremely rare in highly randomized financial asset time series data. Additionally, using FTS2K's KAN network, our predictive results ranked in the top three in all four datasets.

**Real World Treading Result** Figure 3 presents a trend chart of our asset account's performance, as provided by the securities company, showcasing the returns generated by our model in a real trading environment. The chart features a line graph where the yellow line depicts the trend of the CSI 300 of China's A-share market, and the blue line illustrates our investment returns. CSI 300 consists of the 300 stocks, reflecting the overall performance of the China's A-share market. The period for calculating returns spans from January 1, 2024, to October 8, 2024. In our real trading experiment, the FEC-Real model daily recommends 50 stocks based on past trading data, from which we manually select stocks to execute buying and selling instructions. Throughout the investment phase, we continually refined our algorithmic model. Excluding influence of the significant surge in the Chinese stock market since September 24, our investment strategy peaked on May 6, achieving a yield of 52.96%, in contrast to the 6.61% yield of the CSI 300 Index during the same timeframe. However, from June to August, our strategy's returns experienced a more pronounced correction compared to the CSI 300 Index, eventually stabilizing at a 15.35% yield by September 15, while the CSI 300 Index recorded a -6.86% yield in the same period. Despite the trading record covering only nine months, the Chinese stock market in 2024 witnesses various extreme market conditions, including sharp declines, gradual decreases, and significant increases. Although our strategy performs poorly during certain periods, it successfully navigates market challenges, delivering superior returns.

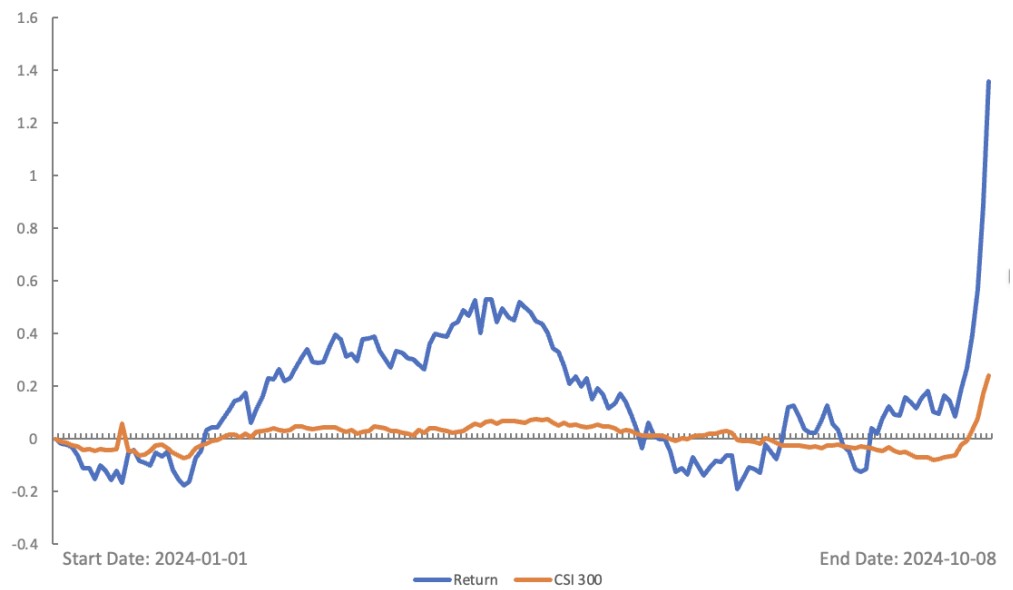

Figure 3: Trend chart of real world transaction return rates.

## 5.1 Ablation Studies

We conduct a series of ablation studies aimed at exploring the impact of various components of the KAN network architecture and the fusion strategies of multiple time series models on the model performance.

**KAN** In the process of experimenting with various configurations of KAN. we aim to maintain the simplicity of the KAN structure, limiting the number of layers to five or fewer, while also investigating the impact of different architectural designs on model performance. We conducted comparisons across a multitude of structural variants, incorporating various numbers of layers and combinations of activation functions, to identify the KAN configuration most suited for financial time series forecasting tasks. As illustrated in the figure 5.1, it showcases the accuracy results across four datasets with different numbers of layers, where the

line graph represents the average accuracy across these datasets. It is evident from the graph that utilizing a three-layer network structure yields the highest average accuracy.

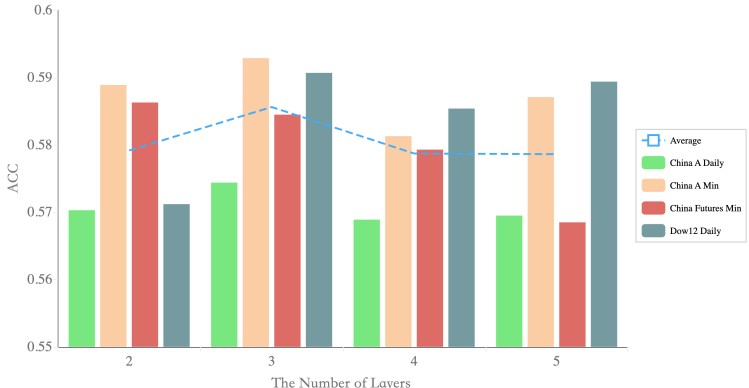

Figure 4: The prediction ACC of the four data sets under different KAN structures.

**Multi-agent**   In our exploration of hybird multiple agents, we investigate various fusion techniques, including: the Hard Vote Method, where each agent independently makes a prediction and the final forecast is determined through majority vote; the Confidence Threshold Method, which considers predictions above a specific confidence level or selects based on confidence level rankings; and the Result Fusion Method, which combines the predictive outcomes of all agents using a Multilayer Perceptron (MLP) network to produce the final prediction. Figure 5 shows the results of three different methods on the experimental dataset. By evaluating these distinct hybird strategies, we can ascertain the most suitable method for our model framework.

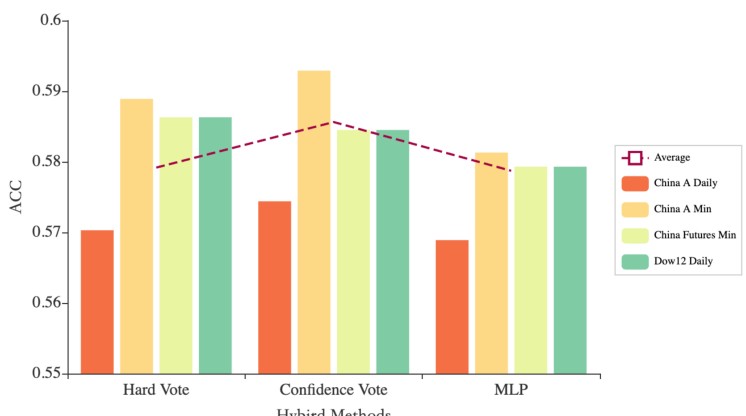

Figure 5: The prediction ACC of the four data sets under different hybrid methods.

**KAN replaces MLP**   We replace the MLP in traditional networks with the KAN, with the goal of determining if this substitution could improve the model's performance or interpretability. The accuracy of using MLP as the Transformer classification network on the China A data set is 0.5664, and the accuracy of using KAN as the classification network is 0.5449, our experiments indicate that directly substituting the MLP with the KAN network leads to a reduction in performance.

**FTS2K Encoder Embedding Dimension**   We used the KAN network to test the performance of the FTS2K encoder in different embedding dimensions on the China A dataset. As shown in Figure 6, when

the output of the FTS2K encoder is 20 dimensions, the prediction ACC of the KAN network is the highest, which is 0.5744.

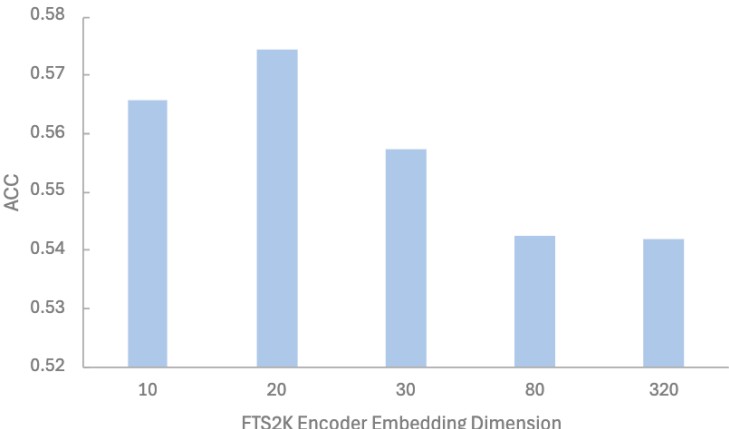

Figure 6: The prediction ACC of the different FTS2K embedding dimensions on the China A dataset.

## 6 CONCLUSION

The research presented in this paper has successfully designed and trained an encoder specifically for financial time series, FTS2K, which can effectively improve the prediction accuracy of various time series models in financial tasks. In addition, we develop and evaluate a hybrid model FEC-Real, which integrates the KAN and FTS2K Encoder for predicting stock price movements. The model demonstrat significant improvements in predictive accuracy over existing state-of-the-art (SOTA) time series models, particularly in handling high-dimensional financial time series data.

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
