# OpenReview forum: "FEC-Real: A KAN based Model for Improving Investment Strategies"
_TMLR — Rejected by TMLR_

### Review · Reviewer_hVnF · 2025-04-06

**Summary Of Contributions:**

The paper presents a hybrid model named FEC-Real, which combines three components: the FTS2K encoder, a KAN-based classifier, and a multi-agent ensemble model. The authors claim that this approach improves the accuracy of stock price prediction by leveraging the strengths of Kolmogorov-Arnold Networks (KAN) and transformer architectures. The key contributions are:

1. FTS2K Encoder: A financial time series encoder designed to capture features from diverse financial data with varying time series lengths. It integrates positional encoding, attention mechanisms, and dimensionality reduction to produce low-dimensional encoded information.
2. KAN as Classifier: The use of KAN, inspired by the Kolmogorov-Arnold representation theorem, to classify financial time series trends. KAN replaces traditional linear weight matrices with learnable univariate functions, aiming to enhance interpretability and efficiency in nonlinear data.
3. Multi-Agent Model: An ensemble approach that aggregates predictions from multiple time series models (KAN, Transformer, TimesNet, PatchTST, Informer) using confidence thresholds and voting mechanisms to improve prediction robustness.

The authors report empirical improvements over several state-of-the-art (SOTA) models, achieving an average accuracy enhancement of 2.62% and demonstrating superior performance on four financial datasets (China A Daily, China A Min, China Futures Min, Dow 12 Daily). Additionally, they highlight real-world trading results, claiming a 46.35% excess return in a real trading environment.

**Audience:**

Yes

**Broader Impact Concerns:**

No mention of ethical risks. Algorithmic trading can destabilize markets or bake in biases. Need a Broader Impact Statement addressing how this model could affect real investors and markets.

**Claims And Evidence:**

No

**Requested Changes:**

1. Expand Ablation Studies:
The ablation studies should investigate the impact of each component (FTS2K encoder, KAN, multi-agent model) in isolation and in combination. This would help isolate the specific contributions of each part of the model and clarify whether the improvements are due to the proposed architecture or the ensemble effect.

2. Clarify Novelty:
The authors should explicitly differentiate their contributions from existing work, particularly in the FTS2K encoder and multi-agent model components. A detailed comparison with related techniques (e.g., transformers, traditional ensemble methods) is necessary to justify the proposed approach. Specifically, how does the FTS2K encoder improve upon standard dimensionality reduction and attention mechanisms? What unique advantages does the multi-agent model offer compared to other ensemble strategies?

3. Cite Recent Work:
The authors should cite and discuss the following recent papers that use KAN for temporal tasks and transformers:

[A] "Kolmogorov-arnold transformer." The Thirteenth International Conference on Learning Representations. 2024.
[B] "A temporal kolmogorov-arnold transformer for time series forecasting." ArXiv (2024).

**Strengths And Weaknesses:**

**Strengths**:

1. The paper addresses a relevant problem in financial time series prediction, which is of significant interest to both academic and industry audiences.
2. The proposed multi-agent ensemble model introduces a practical approach to improving prediction robustness by combining multiple models.
3. The empirical evaluation covers diverse datasets and metrics (ACC, Top-100 ACC), providing a comprehensive assessment of the model's performance.


**Weaknesses**:
1. Lack of Novelty: The FTS2K encoder and multi-agent model appear to repackage existing techniques (e.g., attention mechanisms, dimensionality reduction, ensemble methods) without clear theoretical advancements. The use of KAN, while mathematically grounded, does not extend beyond its original formulation, raising questions about the incremental value over traditional neural networks.
2. Overstated Contributions: The paper presents KAN as a novel and superior architecture for financial time series prediction, but the results don’t convincingly support this claim. Most comparisons are made between an **ensemble model** and **single baseline models**, which isn’t a fair benchmark. The only meaningful comparison is in the section where `KAN replaces MLP`, but even there, performance appears to drop.
3. Methodological Gaps: The integration of KAN with the FTS2K encoder and multi-agent model lacks a clear rationale for why this combination is superior to other hybrid approaches. The ablation studies are limited and do not thoroughly explore the contributions of individual components.

---

> ### Author Response · Authors · 2025-04-26
>
> We would like to thank the reviewer for the detailed comments. Our response is as follows.
>
> **Lack of Novelty** Our contribution is primarily motivated by challenges observed in financial market scenarios. Traditional approaches typically apply predictive models directly to financial data; however, financial market conditions are highly dynamic and prone to rapid shifts. For instance, recent changes in U.S. tariff policies have led to significant alterations in global financial market behavior. In practice, industry models often require frequent updates—sometimes on a monthly, weekly, or even daily basis—to remain effective within the evolving market environment. By employing the FTS2K encoder, we substantially enhance the operational efficiency of prediction models, enabling the use of more streamlined architectures or faster model updates without compromising predictive accuracy.
>
> **Cite Recent Work** We will incorporate the recent work suggested by you in the paper.
>
> **Expand Ablation Studies:**  We have planned to incorporate the following table to show the prediction results of the KAN model alone and the prediction results of the multi-agent.
>
>                                  KAN        | KAN+FTS2K       |   FTS2K+Multi-agent
>
> China A Daily             0.5711           0.5744                        0.5837
>
> Dow12 Daily           0.5814              0.5905                        0.6127
>
> China-A Min            0.5359             0.5929                        0.6072
>
> China-Futures Min  0.5573              0.5845                          0.6198

---

> > ### Comment · Reviewer_hVnF · 2025-06-04
> > **Response to author**
> >
> > Thank the authors for the clarification and additional experiments. However, I still feel that my original concerns have not been fully addressed:
> >
> > - On the lack of novelty: The authors argue that the method is motivated by “challenges observed in financial market scenarios”. However, this does not address my concern. I did not claim that the paper lacks contribution altogether, but rather that the **technical novelty** is limited. Most time-series papers address the problem of markets being “highly dynamic and prone to rapid shifts”. While the proposed method may be effective in practice, the techniques used are well-established and do not offer new insights or innovations.
> >
> > - On the integration of KAN, the FTS2K encoder, and the multi-agent model: I asked for the **rationale and motivation** behind combining these components. Instead of explaining the reasoning or conceptual design, the authors only provided new results. However, **results alone do not explain the motivation** behind the design choices.

---

### Review · Reviewer_KpEF · 2025-04-12

**Summary Of Contributions:**

This paper focuses on the financial application and presents FEC-Real, which integrates KAN and ensemble of multiple forecasting models for final decision-making. The proposed method is effective in China Stoke market.

**Audience:**

Yes

**Claims And Evidence:**

No

**Requested Changes:**

1. Clarify the novelty and provide more implementation details.

2. If it is possible, I think releasing the collected dataset will be valuable to the community.

**Strengths And Weaknesses:**

## Strengths

1. It is interesting to see the effectiveness of KAN in financial decisions.

2. The experimental dataset is quite interesting.

## Weaknesses

1. Lack of technique novelty. In my opinion, the listed three components are just direct applications of well-established techniques, such as the Transformer encoder, KAN, and an ensemble of classifiers. The current design may not be inspiring for this community.

2. Lack of experimental details. (1) What is the target of the proposed method? What do you mean by "positive" and "negative"? Does "positive" refer to that this stock can be increased in the future? By the way, how long will this be in the future?  (2) How to adopt these baselines into a binary classification task? I think most of these baselines are just forecasting models, which are not proposed for classification.

3. How to choose the ensemble forecasters in Section 3.3? Will the performance be consistently improved by adding more forecasting models?

4. Would you please provide some case studies?

---

> ### Author Response · Authors · 2025-04-26
>
> We would like to thank the reviewer for the detailed comments. Our response is as follows.
>
> **Lack of Novelty** We have already addressed this point in reviewer hVnF's comment.
>
> **Lack of Experiment Details** 1) Yes, positive means that the stock price will increase in the future, and negative means that the stock price will decrease in the future. We have also conducted experiments to test the time interval issue you mentioned on China-A dataset. The longer the time interval, the worse the effect. Therefore, we set the prediction time interval to T+2. The reason for this setting is that China's stock trading rules are T+1, so we need to ensure that the price is obtained at the T time point. At the T+1 time point, you can buy at this price and sell at a profit at the T+2 time point. 2) The models in the baseline are indeed mainly prediction models, but these models also provide classification modes (you can view the classification task in the Time-Series-Library project https://github.com/thuml/Time-Series-Library). Since directly predicting prices performs poorly on stocks, we use classification models.
>
> **How to choose the ensemble forecasters in Section 3.3? Will the performance be consistently improved by adding more forecasting models? **  In terms of model selection, we directly selected several models (KAN, Transformer, TimesNet, PatchTST, Informer) with high prediction accuracy. If we add too many models and these models have low accuracy, the result will be worse.
>
> **Dataset Link** If the paper gets published we will release the dataset link. We couldn't upload the dataset in anonymous github due to its large size.

---

### Review · Reviewer_cVUT · 2025-04-24

**Summary Of Contributions:**

This work investigates the use of Kolmogorov Arnold Networks (KANs) for financial time series modeling, in particular, the future trend for stocks. Compared to traditional transformer based models, KANs provide substantial flexibility by using learnable activation functions, which in turn facilitates the simulation of complex functions like the ones in time series analysis. However, the efficacy of KANs is reduced in high-dimensional data. To address this, the authors propose an FTS2K transformer-based encoder, which ultimately produces encoded, compressed information of only 20 dimensions. For improved classification accuracy, their model aggregates the predictions from various time series models by only focusing on the high-confidence forecasts. The extensive experimental evaluation shows that the proposed architecture yields an average accuracy improvement of 2.62% compared to state of the art time series models, while consistently outperforming in four datasets.

**Audience:**

Yes

**Broader Impact Concerns:**

I do not have any particular concerns.

**Claims And Evidence:**

No

**Requested Changes:**

First, I would appreciate it if the authors could respond to my negative comments under the Weaknesses section. Furthermore:

- Given the current scope is narrow, I was wondering whether the authors could broaden it by discussing how their framework can be used for more general time series analysis. Perhaps the focus on stock markets severely limits the target audience of this work.
- The authors should explain what they mean by claiming that "The models in Table 2 directly process the data from the datasets, while the data in Table 3 is encoded by the FTS2K encoder before being fed into the models for prediction". It seems that both Tables 2 and 3 contain options for processing the input with and without FTS2K.
- The x-axis values in Figure 3 are missing, and the authors should add them.
- Since FEC-real consists of 3 components (encoder + KAN + multiagent), I believe it would help if the authors clarified the improvement from each of the three components. Actually, Tables 2 and 3 kind of do that, but it is not as clear in the current presentation.
- The authors should provide some background knowledge on financial time series forecasting, like how exactly the trend is defined. Furthermore, what is the exact classification task, and could this framework also be used with regression tasks?

**Strengths And Weaknesses:**

Strengths
- The paper proposes an interesting combination of transformer-based encoder with a KAN network, which resolves the efficacy problem of traditional KANs under high-dimensional data.
- The work is well motivated: KANs make sense for time series modeling because they can model complex functions via activation function learning, while the FTS2K encoder performs dimensionality reduction to make KAN more efficient.
- The model aggregation technique seems to work well, even if it is not a novel idea.
- The experimental analysis is quite extensive, since the authors compare to a wide array of previous, state of the art approaches, including the more recent ones. Also, the authors use several datasets (4). The improvement in average accuracy is pronounced and consistent across all datasets, which confirms the value of the proposed architecture.
- The experimental results show that the framework works well both with high-dimensional as well as low-dimensional datasets, although the gain is more pronounced in the latter.

Weaknesses
- The focus of this work is very narrow, making it relevant only to a very small part of the AI/ML community. Essentially, this proposed architecture could work for general time series modeling, eve outside the stock/finance market. It seems to me that the authors would want to make the paper scope as wide as possible, but currently it focuses almost exclusively on financial time series. I understand that such time series are inherently complex - but why not focus on the more general setting?
- I was confused by Tables 2 and 3 in Section 5. The authors claim that "The models in Table 2 directly process the data from the datasets, while the data in Table 3 is encoded by the FTS2K encoder before being fed into the models for prediction." But in both Tables 2 and 3, there are columns with an additional letter 'E', indicating that "the dataset has been encoded by FTS2K". Isn't there a contradiction here? It seems that both Tables allow for processing with and without FTS2K, and the only difference is the datasets: Table 2 focuses on low-dimensional datasets, whereas Table 3 on high-dimensional ones. I may have misunderstood this point, could the authors make it a lot clearer?
- I have some concerns regarding Figure 3. First, notice that the dates on the x-axis are not depicted. This creates presentation problems, because the reader has no way of locating the dates "September 24", "September 15" etc., that are mentioned in the main text. Second, it is true that the proposed method achieves a remarkable return in the first half - but it can also dip to lower values, compared to the CSI 300 curve. I am not sure what that says about the robustness of the proposed framework. Is it prone to riskier behavior?
- The authors claim to use a multi-agent model that aggregates the predictions from various models (KAN, Transformer, TimesNet, PatchTST, Informer). To me, it would have been important to show how the proposed framework performs without this multi-agent module, i.e., if it only uses its own prediction without integrating predictions of other models. This would show what is the actual gain achieved by the proposed module. Do the authors conduct such an analysis in the paper? My understanding is that this is exactly the column 'KAN' in Tables 2 and 3, i.e., they show how the KAN network performs without the multiagent component. But if we look at China-Futures E, then Transformer performs better than KAN. So, it seems the benefit for KAN is not as large as the authors claim.
- The authors claim that their goal is to classify the trend of financial time series data. How exactly is this a classification problem? They state that there are two output nodes to judge the future trend of the financial time series. Is that a binary classification problem "up" or "down"? Or is it something else? I feel it would help if the authors explained their problem statement better, especially for people who are not familiar with the finance literature.

---

> ### Author Response · Authors · 2025-05-06
>
> We would like to thank the reviewer or the detailed comments. Our response is as follows.
>
> **Confusion between Table 2 and Table 3** We sincerely apologize for the confusion caused by this issue, which resulted from an oversight on our part. In the initial version of the paper, the results in Table 2 were presented without utilizing the encoder, while those in Table 3 incorporated the encoder's implementation. Subsequently, to enable clearer experimental comparisons, we revised Table 2 to focus on low-dimensional data and Table 3 on high-dimensional data. We will address this issue in subsequent revisions of the paper.
>
> **Figure 3 Concerns** Thank you for highlighting these two temporal points. We will revise Figure 3 to include annotations for these time points to improve readability. Regarding the robustness concerns, we will evaluate them in the further portfolio management task using the Sharpe ratio and maximum drawdown metrics. The goal of the time-series task is to assess each model's ability to perceive financial time-series data through the accuracy of future trend predictions, which will then serve as the foundation for building a reinforcement learning-based portfolio management agent.
>
> **Clarifications on Up and Down Definition** "up" means that the stock price will increase in the future, and "down" means that the stock price will decrease in the future.
>
> **Ablation Study for KAN** We have now one some additional experiments to show the effectiveness of KAN
>
>
>
>                           KAN    KAN+FTS2K       FTS2K+Multi-agent      FTS2K+Multi-agent (without KAN)
>
> China A Daily   0.5711     0.5744                  0.5837                               0.5798
>
> Dow12 Daily     0.5814    0.5905                   0.6127                               0.5993
>
> China-A Min      0.5359    0.5929                  0.6072                               0.5846
>
> China-Futures Min 0.5573  0.5845               0.6198                                 0.5819

---

### Decision · Action_Editor_CpWm · 2025-06-10

**Recommendation:** Reject

**Audience:**

No

**Audience Explanation:**

The current version is not self-standing.

**Claims And Evidence:**

No

**Claims Explanation:**

The review team including three reviewers and I, reach the consensus rejection recommendation. I checked all the comments from three reviewers and did not notice any unqualified comments. Instead of repeating their opinions, I post my comments below.

1. Motivation. The motivation of this paper is unclear. The authors argue that "transformer based models are not always efficient for real world high speed trading data" in the abstract and "given the challenge of balancing prediction accuracy and generalization within a single model, researchers have started to investigate hybrid techniques" in the introduction part. Everyone should agree that there is no algorithm that can always work; therefore, the first motivation does not make sense to me. For the second one, since there exists the hybrid solitons in the literature, it is unclear why the authors would like to propose another one. In another word, the authors should illustrate the drawbacks of existing solutions. Unfortunately, it is missing.

2. Technique. The authors propose a hybrid model that combines transformer and KAN. Since the motivation is missing in my eyes, it is prohibitive for me to judge whether the proposed techniques can tackle the target challenging, which is missing.

3. Experiments. Although the authors provide the comparative results of algorithmic performance, they fail to verify their motivations, i.e., the efficiency of the proposed model and the balance of predictive accuracy and generalization.

4. Presentation. Some citation formats are incorrect. Pay attention to the difference among \cite, \citet, and \citep. Moreover, there should have a space before the citations.

Therefore, the claims in this paper is not sufficiently supported and does not meet the standard of TMLR.

Why a rejection not the major revision?

The motivation serves as the fundamental backbone of a paper. If this part is not well-grounded, then the overall research philosophy (which I did not see articulated here), the technical solutions, and the empirical evaluations built upon it all become unconvincing. If the authors decide to revise the motivation, then everything derived from it should be reconsidered and potentially reconstructed accordingly.